# Preliminary Study on Expression and Function of the Chicken W Chromosome Gene *MIER3* in Embryonic Gonads

**DOI:** 10.3390/ijms24108891

**Published:** 2023-05-17

**Authors:** Xiao Lin, Zidi Jin, Shuo Li, Mingde Zheng, Ya Xing, Xikui Liu, Mengqing Lv, Minmeng Zhao, Tuoyu Geng, Daoqing Gong, Debiao Zhao, Long Liu

**Affiliations:** College of Animal Science and Technology, Yangzhou University, Yangzhou 225009, China

**Keywords:** chicken embryo, gonadal development, *MIER3*, *αGSU*

## Abstract

The sex chromosomes of birds are designated Z and W. The male is homogamous (ZZ), and the female is heterogamous (ZW). The chicken W chromosome is a degenerate version of the Z chromosome and harbors only 28 protein-coding genes. We studied the expression pattern of the W chromosome gene *MIER3* (showing differential expression during gonadogenesis) in chicken embryonic gonads and its potential role in gonadal development. The W copy of *MIER3* (*MIER3–W*) shows a gonad-biased expression in chicken embryonic tissues which was different from its Z copy. The overall expression of *MIER3–W* and *MIER3–Z* mRNA and protein is correlated with the gonadal phenotype being higher in female gonads than in male gonads or female-to-male sex-reversed gonads. Chicken MIER3 protein is highly expressed in the nucleus, with relatively lower expression in the cytoplasm. Overexpression of *MIER3–W* in male gonad cells suggested its effect on the GnRH signaling pathway, cell proliferation, and cell apoptosis. *MIER3* expression is associated with the gonadal phenotype. *MIER3* may promote female gonadal development by regulating *EGR1* and *αGSU* genes. These findings enrich our knowledge of chicken W chromosome genes and support a more systematic and in-depth understanding of gonadal development in chickens.

## 1. Introduction

Chickens are economically important model animals. An in-depth understanding of the mechanism of sex determination and gonadal development in chickens has both theoretical and practical applications. In poultry production, it will be of great value to the poultry industry if the sex determination mechanism can be used to control the gender of hatched chicks which is naturally half male and half female. Differences in sex chromosomes between males and females form the genetic basis for sex determination. Similar to other birds, chickens have a Z/W sex chromosome system in which males are homozygous (ZZ) and females heterozygous (ZW). The Z chromosome is a large chromosome with >1000 genes, while the W chromosome is its degenerate copy containing only 28 protein-coding genes [1,2]. Due to the lack of global dosage compensation, such as the X chromosome in mammals, most Z chromosome genes in chickens have a dosage difference between males and females [3]. According to the most recent build of the chicken W chromosome (GRCg7b), all the protein-coding W genes have their homologous copies on the Z chromosome. These Z-W homologues are thought to have important roles in development [2]. Although some of the W genes were reported to be related to early sex differentiation [4], the Z chromosome dosage effect is a widely accepted mechanism for sex determination in chickens [5,6]. A recent study has supported this view. The heterozygous knockout of the Z chromosome gene *DMRT1* reportedly allows the complete development of ovaries in ZZ-type chicken embryos, indicating that *DMRT1* is a good candidate gene for chicken sex determination [7,8]. However, these Z^+^Z^−^ individuals did not develop into functional females (unable to lay eggs) when they reached adulthood and remained male in appearance and body type [7]. This suggests that chickens may have more complex mechanisms of gonadal development (especially for females) other than the sole control of Z chromosomal genes’ dosage effect, such as the synergistic effects of W chromosomal genes. Therefore, in our previous study, we examined the expression profiles of the 28 W chromosome protein-coding genes and their Z chromosome homologs in chicken embryonic gonads to screen for genes that show differential expression between females and males and identify genes that may be involved in gonadal development in chickens.

Here we studied one of the W chromosome genes *MIER3* (mesoderm-induced early response protein family member 3), which is a transcription factor [9]. In our preliminary results, *MIER3* showed spatiotemporal expression specificity (W chromosome copy is highly expressed in the gonads, and there are differences in overall gene expression between males and females). So, we suggested that *MIER3* might be involved in the development of chicken gonads. To test this hypothesis, we investigated the detailed expression pattern of *MIER3* in chicken embryonic tissues (mainly the gonads). Additionally, we explored its potential function in embryonic gonadal development using a fadrozole (FAD, an aromatase inhibitor which can cause female-to-male sex reversal) induced sex reversal model and analyzed the downstream pathways by *MIER3* overexpression in male gonadal cells.

## 2. Results

### 2.1. Characterization of mRNA and Protein Sequence of Chicken MIER3

Sequences encoding chicken *MIER3* present on the Z and W sex chromosomes were designated *MIER3–Z* and *MIER3–W*, respectively. According to the latest version of the Ensembl database (GRCg7b), there are seven predicted transcripts for *MIER3–Z* and *MIER3–W* in chickens, respectively; the comparison of gene structures of different transcripts is shown in Figure 1A. Overall, *MIER3–W* and *MIER3–Z* had similar gene structures at the 3′ end, and different exon combinations were generated at the 5′ end. Figure 1B and Appendix A show the protein sequence comparison between MIER3–W, MIER3–Z, and human MIER3 (isoform 1, NM_001297598.2). Compared to human MIER3 (isoform 1, encoding 555 amino acids), all chicken MIER3 amino acid sequences had a five-amino-acid fragment deletion at the N-terminus. Besides the fragment deletion, all MIER3–Z amino acid sequences encoded proteins were similar or shorter than those in humans. Multiple transcripts of MIER3–*W* (1,2,3,7) had an additional segment of the amino acid sequence in the front region. Structural domain analysis revealed that human MIER3 (isoform 1), MIER3–W1, and MIER3–Z1 have three conserved domains: SANT, SANT-MTA3-like, and ELM2. The SANT domain protein plays a central role in chromatin recombination. It acts as the only histone tail display module to couple histones with enzyme catalysis. Further analysis showed that almost all MIER3 proteins (except MIER3–W3 and MIER3–*Z3*) contained a conserved SANT-MTA3-like domain in the central part of the open reading frame (cd11661, marked in Figure 1B).

Furthermore, we analyzed the phylogenetic relationships between chicken *MIER3–W*, *MIER3–Z*, and the amino acid sequences of *MIER3* from other animals (all using the amino acid sequence of the longest transcript in the NCBI database), including human (XP_011541519.1), mouse (NP_766181.2), bovine (NP_001291684.1), house swallow (XP_039946523.1), domestic goose (XP_047905371.1), and other 15 species (a total of 21 species, detailed in Appendix A). The phylogenetic tree consisted of two main branches, one consisting of African lungfish (XP_043919732.1) alone and the second, other mammals, reptiles, birds, and amphibians (Appendix A). Notably, chicken MIER3–W and MIER3–Z are not in the closest branches.

### 2.2. Spatiotemporal Expression of Chicken MIER3 in Embryonic Tissues

Different sets of specific primers were designed for *MIER3–W* (a region common to all seven W transcripts), *MIER3–Z* (common to all seven Z transcripts), and *MIER3–ZW* (common to both transcripts) for qPCR quantification of expression in male and female gonadal and non-gonadal (tail) tissues of E6 (day 6 of incubation, before sex determination) and E12 (after sex determination, during gonadal development) embryos. The positions of these primers on the individual transcripts are shown in Figure 1A, and the detailed sequences are provided in Appendix A. As expected, *MIER3–W* (Figure 2A) was expressed only in female tissues, whereas no amplification was detected in male tissues. On E6 and E12, *MIER3–W* expression was higher in female gonads than in tail tissues; in contrast, *MIER3–Z* expression was similar between gonads and tails and was higher in males than in females (except in E12 gonads). Additionally, *MIER3–Z* expression was significantly higher in E6 tissues than in E12 tissues (Figure 2B); *MIER3–ZW* expression was significantly higher in gonads than in tail tissues, and higher in E6 female gonads than in E12 female gonads (Figure 2C). *MIER3–W* expression was significantly higher than *MIER3–Z* in E6 and E12 female gonads (Figure 2D). To clarify the expression of different transcripts of *MIER3–W*, we designed primers to amplify different transcripts according to their unique sequence (Appendix A) and performed the assay of amplification efficiency (Appendix A), and the assay of expression of different transcripts (Appendix A) in E12 female gonads by qPCR, and the results showed that transcripts 3–6 were relatively highly expressed, while transcripts 1, 2, and 7 were very lowly expressed. In addition, since the sum of all known transcripts’ expressions is much lower than the total *MIER3–W* expression, it is possible that other unknown W transcripts are also present.

We further determined *MIER3* expression in other tissues (gonads, mesonephros, leg muscle, cerebrum, glandular stomach, and intestine in E6; gonads, mesonephros, liver, spleen, kidney, cerebrum, breast muscle, leg muscle, heart, glandular stomach, gizzard, and intestine in E12). We found that *MIER3–W* expression was significantly higher in the gonads of E6 and E12 embryos than in other tissues (Appendix A). Moreover, *MIER3–Z* expression levels varied greatly in different tissues and were higher in males than in females in many tissues (Appendix A). In E6 and E12 embryos, *MIER3–ZW* was expressed at significantly higher levels in females than in males in a fraction of tissues (Appendix A). To further investigate the differential expression of *MIER3–Z* in the left-side gonads of female and male chicken embryos, we examined the expression of *MIER3–Z* in the left-side gonads of E6 and E12 female and male chicken embryos. The results (Appendix A) showed that the expression of *MIER3–Z* in the left-side gonad of male chicken embryos was higher than that of female chicken embryos (1.4-fold and 1.3-fold).

A series of protein blot analyses were performed on protein samples collected from male and female tissues from different embryonic stages (E6 and E12; Figure 2E,F). Since the predicted AA sequence of chicken MIER3–W was similar to MIER3–Z and had the same antibody binding region (Figure 1B), our western blot analysis may represent MIER3–Z and MIER3–W combined protein expression. Furthermore, consistent with the higher combined RNA transcript levels (*MIER3–ZW*) in female gonads, *MIER3* levels were higher in female gonads than in male gonads in E6 and E12 (Figure 2E,F).

### 2.3. Localization of MIER3 Protein in Chicken Embryonic Gonads

Immunostaining was used to determine the cellular localization of *MIER3* protein in male and female embryonic gonads at different developmental stages (E12 and Day 1). Staining revealed MIER3 protein presence in male and female gonads (Figure 3A). Additionally, the sections were co-stained with Hoechst (a cell nuclear marker). Immunostaining showed a clear morphological difference between female gonads (ovaries) and male gonads (testes) had emerged at E12. At this point, the cortex and medulla were present in female gonads; in males, the cortex had degenerated, and sex cords had formed in the medulla. The MIER3 protein was expressed in the medulla and cortex of females at E12 but was highly expressed in the cortical layer. On Day 1, MIER3 was expressed mainly in the medulla of female gonads, with low expression in the cortex; however, it was widely expressed throughout the male gonads at E12 and Day 1. Further observation with higher magnification revealed (Figure 3B) that MIER3 protein was present in the nucleus and cytoplasm of ovarian and testicular gonad cells and was present intracellularly and extracellularly in the male gonads. Western blot analysis of proteins from the cytoplasmic and nuclear fractions isolated from E12 female gonads (Figure 3C) confirmed that chicken MIER3 protein was mainly highly expressed in the nucleus, with relatively lower expression in the cytoplasm.

### 2.4. MIER3 Expression in FAD-Treated (Sex Reversal) Embryonic Gonads

As chicken *MIER3* is expressed at different levels in female and male gonads (Figure 2), we evaluated the effect of sex reversal (ovary to testis) to confirm further whether *MIER3* expression correlates with the gonadal phenotype. Figure 4A shows the gross morphology of E12 gonads in each treatment group (male and female gonads in the PBS-injected control group and male and female gonads in the FAD-treated group), indicating that sexual reversal occurred. Quantitative PCR analysis revealed that *MIER3–W* expression was significantly lower in the gonads of FAD-treated females than in the control females (Figure 4B). Additionally, FAD-treated female gonads showed no significant change in *MIER3–Z* expression (Figure 4C); however, *MIER3–ZW* was significantly reduced in the gonads of FAD-treated females (Figure 4D). FAD treatment affected the MIER3 protein levels, as MIER3 protein levels in the FAD-treated female gonads were lower than that in the female control gonads (Figure 4E). Immunostaining showed that in sex-reversed female embryos, the cortex thinned, and sex cords appeared inside the medulla. Additionally, MIER3 changed from being highly expressed in the cortex to uniformly expressed in the medulla. However, FAD injection had little effect on the internal structure of the gonads of male embryos and MIER3 expression in gonadal cells (Figure 4F). These analyses suggest that *MIER3* (transcript and protein) expression is associated with the gonadal phenotype.

### 2.5. Downstream Genes and Pathways of MIER3 in Gonadal Cells

To investigate the downstream genes and pathways of the *MIER3* gene, we overexpressed the *MIER3–W* gene in the primary gonadal cells of E9.5 male chicken embryos. Quantitative PCR and western blot analysis showed that the expression of *MIER3* mRNA and protein was significantly higher in gonadal cells transfected with *MIER3* overexpression vector than in those transfected with empty vector (Figure 5A,B). Additionally, transcriptome sequencing of male gonadal cells transfected with *MIER3* overexpression vector or empty vector showed that 75 DEGs (Differentially Expressed Gene) were identified in the *MIER3* gene overexpressed cells compared to control cells (30 upregulated and 45 downregulated, Appendix A). The top 10 upregulated and downregulated genes with the lowest *p*-values are shown in Appendix A, respectively.

To verify the reliability of the transcriptome analysis, we selected 10 DEGs for qPCR validation (Appendix A), including five upregulated genes, namely C-C motif chemokine ligand 4, cell cycle protein-dependent kinase inhibitor 2A, cell cycle protein-dependent kinase inhibitor 2B, complexing agent 1, and prostaglandin-endoperoxide synthase 2; five downregulated genes, namely *EGR1* (early growth response 1), tumor necrosis factor receptor superfamily member 13B, G protein signaling regulator 1, glutathione-specific γ-glutamyl cyclotransferase 1, and *FREM1* (Fraser extracellular matrix complex subunit 1-related fine extracellular matrix 1). The data showed that 9 out of 10 genes were validated (except *FREM1*), indicating that the transcriptome analysis was generally reliable (Figure 5C). GO enrichment analysis (Appendix A) of DEGs showed that the significantly enriched GO terms included positive regulation of melanosome organization during endothelial cell differentiation, notch signaling pathway, pigment granule organization, α-amino acid metabolic processes, and carbon and nitrogen cleavage enzyme activity (Appendix A). KEGG pathway analysis (Figure 5D) revealed a significant enrichment of DEGs in cytokine–cytokine receptor interactions, the gonadotropin-releasing hormone (GnRH) signaling pathway, cell adhesion molecules, and histidine metabolism (Figure 5D).

Additionally, since *MIER3* overexpression affects the GnRH signaling pathway, we further investigated gonadotropin (*αGSU*), a gene downstream of *EGR1* (previously validated by qPCR) in this pathway. Transcriptome sequencing showed that *αGSU* expression was elevated, but not significantly, in the overexpression group (Appendix A, n = 3, *p* = 0.79, FDC = 1.3). Furthermore, qPCR validation revealed that *MIER3–W* overexpression promoted *αGSU* expression (Figure 5F, n = 5, *p* = 0.02, FDC = 1.9). Consistent with this result, *αGSU* expression was significantly higher in the E6 female gonads than in the male gonads (Figure 5E). However, these results contradicted the downregulation of *EGR1* in the overexpression group (*EGR1* promotes *αGSU* expression in the GnRH signaling pathway), and we speculated that there might be post-transcriptional regulation of *EGR1* in this process. Further western blot studies confirmed this prediction, showing that the protein level of EGR1 was significantly increased after *MIER3* gene overexpression (Figure 5G).

### 2.6. Effect of MIER3 Gene on Cell Phenotype

Furthermore, because *MIER3* overexpression affected cytokine–cytokine receptor interactions, the apelin signaling pathway, the ErbB signaling pathway, and other pathways related to cell phenotype, we verified the effect of *MIER3* overexpression on cell phenotype. After *MIER3* overexpression in gonadal cells, male cells proliferated significantly higher at 12 h, 24 h, and 72 h post-transfection (Figure 6A) and female cells at 72 h and 96 h (Figure 6B). In contrast, *MIER3* gene overexpression in female and male gonads inhibited apoptosis (Figure 6C,D) and increased the distribution of cells in the G2 phase of the cell cycle in female gonads (Figure 6E).

## 3. Discussion

Sex determination and gonadal development is a delicate and complex process controlled by multiple genes. Birds have Z and W sex chromosomes, with males being homozygous (ZZ) and females being heterozygous (ZW), different from the XY chromosome system in mammals, such as humans. Correspondingly, sex determination mechanisms in birds are different from those in mammals. To date, no sex-determining genes similar to *SRY* (sex-determining region Y) on the human Y chromosome have been identified in chickens [10]. Although the *DMRT1* gene on the chicken Z chromosome is considered the most likely sex-determining gene [11], female gonadal development seems to have a more complicated mechanism, which involves multiple pathways (e.g., *FOXL2*-*AROM*-*ERα*, *RSPO1*-*WNT4*, and maybe unverified W genes) [7,12,13]. Theoretically, genes on the W chromosome are likely to regulate female gonadal development in chickens.

Consequently, we explored the possibility that one of the genes on the W chromosome, *MIER3*, is involved in female gonadal development in chickens. We found that *MIER3* gene expression in gonads at both the mRNA and protein levels and its protein distribution in gonadal tissues correlated with the gonadal phenotype. Further studies revealed that *MIER3* gene might promote female gonad development through *EGR1* and *αGSU* genes. These findings suggest that the *MIER3* genes are involved in gonadal development in chicken embryos, increasing our understanding of the role of chicken W chromosome genes in gonadal development.

To predict the potential functions of the chicken *MIER3* gene, we first performed a comparative analysis of the mRNA and amino acid sequences encoded by different transcripts of the Z and W copies of the chicken *MIER3* gene and the human *MIER3* transcript 1. Notably, owing to evolutionary differences, unlike the chicken *MIER3* gene, human *MIER3* is not on the sex chromosome (located on autosome 5). Nevertheless, the amino acid sequences of human MIER3 (isoform 1) and chicken MIER3 (especially the Z-copy isoform) are generally similar, and both have three conserved structural domains, namely SANT, SANT-MTA3-like, and ELM2 [14,15,16,17], suggesting that chicken MIER3 may have functions similar to those of human MIER3. A study on human MIER3 showed that NAT9 (N-acetyltransferase 9) is a candidate for interacting with MIER3 [18]. Breast cancer studies have revealed that *MIER3* is highly expressed in breast cancer, suggesting that human *MIER3* is closely associated with the development of breast cancer [19]. Furthermore, human *MIER3* is reportedly lowly expressed in colorectal cancer cells, which may correlate with the degree of tumor differentiation, suggesting that *MIER3* is closely associated with colorectal carcinogenesis [2]. However, no study has reported the function of chicken *MIER3*. Notably, compared with human *MIER3*, multiple transcripts of *MIER3–W* (1,2,3,7) have an extra segment of amino acid sequence at the N-terminus (Figure 1), and the phylogenetic tree revealed (Appendix A) that chicken *MIER3–W* and *MIER3–Z* are not in the closest branch, suggesting that *MIER3–W* may have some specific functions.

Additional evidence comes from the specific expression pattern of *MIER-W* in the chicken embryonic gonads. *MIER3–W* was significantly more expressed in chicken embryonic gonads than in other measured tissues (Figure 2A and Appendix A); *MIER3–Z* was not (Figure 2B and Appendix A), suggesting that *MIER3–W* may have certain functions related to gonadal development. Furthermore, *MIER3–W* expression was significantly higher than *MIER3–Z* in E6 and E12 chicken embryo gonads (Figure 2D), suggesting that *MIER3–W* is not just compensation for *MIER3–Z*. This result is in line with the previous RNA-seq data of the W genes and contradicts the previous report that the W chromosome gene compensates for the Z chromosome copy [2]. However, consistent with previous reports, there was a dosage effect of *MIER3–Z* between males and females in most embryonic tissues [2]. For example, *MIER3–Z* was higher in males than in females in both E6 and E12 gonadal and tail tissues (Figure 1B) and was also higher in males than in females in multiple non-gonadal tissues (E6: cerebrum; E12: liver, kidney, cerebrum, leg muscle, glandular stomach, and intestine).

Notably, in the gonads of FAD-treated female embryos, the mRNA expression levels of both *MIER3–W* and *MIER3–ZW* were reduced, and a significant reduction was observed in protein levels. Additionally, FAD treatment affected MIER3 protein distribution in gonadal cells (from high expression in the female cortex to uniform expression in the medulla), suggesting that the *MIER3* gene is closely related to the gonadal phenotype.

Furthermore, we used IF (Immunofluorescence) to determine the cellular localization of MIER3 protein in male and female embryonic gonads at different developmental stages (E12 and Day 1). *MIER3* gene had different distribution patterns of protein localization in the female and male gonads. In E12 embryos, MIER3 protein was highly expressed in the cortical layer of female gonads and uniformly expressed in the medullary layer of male gonads; in Day 1, it was highly expressed in the medullary layer of female gonads, with very low expression in the cortical layer. Since it has been demonstrated that *MIER3* can affect cell proliferation [20], its high expression in the cortex of female gonads at early stages may contribute to the maintenance of the female cortex. *TGIF1* may be involved in this process, as it was identified as a regulator of proximal cortical medulla development [21]. Our transcriptome sequencing results showed that *MIER3* overexpression increased *TGIF1* gene expression (Appendix A). Furthermore, MIER3 protein is expressed in the nucleus and cytoplasm, with high levels in the nucleus, which is consistent with the results of Roya Derwish and Man Peng [17,22], proving that MIER3 is a nuclear protein. However, the MIER3 protein was expressed intracellularly and extracellularly, which was quite obvious in male gonads (Figure 3B), suggesting that MIER3 may also function as a secretory protein.

To explore *MIER3*–related signaling pathways and genes, we overexpressed *MIER3–W* in E9.5 male gonadal cells. With transcriptome sequencing, we screened a series of differential genes and signaling pathways, especially the GnRH signaling pathway (GnRH secreted by the hypothalamus acts on receptors in the anterior pituitary to regulate *αGSU*, *LH*, and *FSHβ* production and release). We found that *MIER3* overexpression can significantly increase EGR1 protein and *αGSU* (a gene downstream of *EGR1*) mRNA expression. Furthermore, *αGSU* was significantly higher in female gonads than in male gonads at E6, suggesting that the *MIER3* gene may promote the development of female gonads through *EGR1* and *αGSU*. However, there was a contradiction between the mRNA and EGR1 protein levels in the gonadal cell’s *MIER3* overexpression, indicating the existence of post-transcriptional regulation of *EGR1*. The detailed mechanism of this process warrants further study. In addition, we noticed that the overexpression of *MIER3–W* had no effect on E2 pathway genes (*FOXL2*, *AROM,* and *ERα*). Given the fact that FAD treatment (aromatase inhibition) decreased the expression of *MIER3*, it is suggested that *MIER3* expression is downstream of the E2 pathway.

## 4. Materials and Methods

### 4.1. Animal Ethics Statement

All animal protocols were approved by the animal welfare committee of the Yangzhou University [permission number: SYXK(Su) IACUC 2012–0029] and comply with the associated guidelines.

### 4.2. Animal and Sample Collection

The eggs were all obtained from Hyline grey laying hens from Conway Farm (Yangzhou, China) and incubated in the laboratory of Yangzhou University. They were incubated at 37.5 °C with 60% humidity, blunt end up, and rotated every 30 min until the desired embryonic stages. On E6 (n = 6), E12 (n = 6), and hatch (gonadal morphology fully developed, n = 6), the eggs were removed from the incubator, and the embryos were carefully dissected to expose and collect the gonads and other tissues. The gonads and tails of E6 and E12 embryos were collected into 1.5 mL Eppendorf tubes for RNA extraction with TRIzol (Cat. No. DP424; Tiangen Biotech Co., Ltd., Beijing, China) and 200 μL of RIPA Lysis buffer (Cat. No. C1053; Applygen Technologies Co., Ltd., Beijing, China) for protein extraction. The complex renal tissues (gonad and mesonephros) were collected from the gonads of E12 and hatch, fixed in 4% paraformaldehyde, and subjected to paraffin sectioning and immunostaining.

### 4.3. Building a Phylogenetic Tree

Using MEGA 5.05 software, phylogenetic trees of amino acid sequences of *MIER3* genes from different animals were constructed using the adjacency method, including: chicken (XP_004937308.1, XP_015155926.1), homo sapiens (XP_011541519.1), Mus musculus (NP_766181.2), Bos taurus (NP_001291684.1), Sus scrofa (XP_013840248.1), Canis lupus familiaris (XP_038315053.1), Equus caballus (XP_023481416.1), Felis catus (XP_023112535.1), Oryctolagus cuniculus (XP_008260452.1), Chelonia mydas (XP_037758759.1), Protopterus annectens (XP_043919732.1), Trachemys scripta elegans (XP_034630109.1) Lagopus leucura (XP_042731061.1), Cygnus olor (XP_0403955799.1), Hirundo rustica (XP_039946523.1), Oxyura jamaicensis (XP_035166467.1), Zootoca vivipara (XP_034956846.1), Phocoena sinus (XP_032484360.1), Aythya fuligula (XP_032062173.1), Anas platyrhynchos (XP_038025982.1), and anser cygnoides (XP_047905371.1).

### 4.4. FAD Treatment of Chicken Embryos

E2.5 (Embryonic day 2.5) embryos were injected with FAD, an aromatase inhibitor that causes gonadal sex reversal in female embryos [23]. A small hole was made in the blunt end of the egg, and 0.2 mg FAD dissolved in phosphate buffer solution (PBS) was injected into the air sac. The eggs were then sealed and re-incubated until E12. Next, gonad samples were collected for RNA (n = 6) or protein (n = 6) extraction, as described above.

### 4.5. Determination of Genetic Sex of Chicken Embryos

DNA was extracted from chicken embryonic tissues (wings or toes) using a commercial kit (Cat. No. DC102-01; Novozyme Biotechnology Co., Ltd., Nanjing, China). Subsequently, PCR was performed to amplify the *CHD* gene (Chromobox-helicase-DNA binding gene) sequence located on both sex chromosomes. The *CHD*-forward/reward primer sequences were as follows: *CHD*-F, AGTGCATTGCAGAAGCAATATT; *CHD*-R, GCCTCCTGTTTATTATAGAATTCAT. The female (ZW) had two bands at 506 bp and 351 bp, and the male (ZZ) had only one band at 506 bp.

### 4.6. RNA Isolation, cDNA Synthesis, and qPCR

Total RNA was extracted from the gonads and other tissues using TRIzol universal total RNA extraction reagent (Cat. No. DP424; Tiangen Biotech Co., Ltd.) according to the manufacturer’s instructions. For embryos at E6, five pairs of gonads were pooled for RNA isolation (6 pools for each sex), and for embryos at other stages (E12 and hatch) and in cell experiments, total RNA was extracted from six pairs of gonads of each sex (n = 6). The HiScript Q RT SuperMix for qPCR reverse transcription kit (Cat. No. R123-01; Novozyme Biotechnology Co., Ltd., Nanjing, China) was used for cDNA synthesis. Quantitative PCR was performed using a VazymeAceQ qPCR SYBR Green Master Mix kit (Cat. No. Q111- 02/03; Vazyme Biotech Co., Ltd., Nanjing, China), according to the manufacturer’s instructions. The hydroxymethylbilane synthase gene was used as a control for the spatiotemporal expression of the *MIER3* gene in chicken embryos [24], and the actin (*β-actin*) gene was used as a control for the rest of the assays. The primer sequences used for qPCR analysis are listed in Appendix A.

### 4.7. Protein Extraction and Western Blotting

Nuclear and cytoplasmic protein fractions were prepared from E12 female gonads (n = 3) using the “NE-PER Nucleus and Cytoplasm Extraction Reagent” kit (Cat. No. 78833; Thermo Scientific Co., Ltd., Boston, MA, USA), according to the manufacturer’s instructions. Total protein from the gonads was extracted using RIPA buffer (Cat. No. C1053; Applygen Technologies Co., Ltd., Beijing, China) according to the manufacturer’s instructions, and protein concentrations in the samples were determined using the Enhanced BCA Protein Analysis Kit (Cat. No. P0010; Beyotime, Shanghai, China). Immunoblot analysis was performed as described previously [25]. Briefly, 4 μg of protein sample was subjected to sodium dodecyl sulfate-polyacrylamide gel electrophoresis and transferred to PVDF (polyvinylidene fluoride) membranes. After sealing, the membranes were incubated with the primary antibody overnight at 4°C and then incubated with secondary antibody for 1 h at room temperature (15–25 °C) before imaging. Scanned images were analyzed using Image Studio Lite Ver 5.2. MIER3 antibody (bs-49157R) (Boosun Biotechnology Co., Ltd., Beijing, China), EGR1 antibody (ab133695, Abcam, Cambridge, UK), and GAPDH antibody (bs-0755R, Boosun Biotechnology Co., Ltd., Beijing, China) were used as a control.

### 4.8. Paraffin Sectioning and Immunostaining

For histological analysis, the complex renal tissues were placed in 4% paraformaldehyde for 24 h. Next, the tissue shape was adjusted under a stereomicroscope. Subsequently, the tissue and corresponding labels were placed in a dehydration cassette, dehydrated with low to high ethanol concentrations, treated with xylene, and cleared for paraffin embedding. The embedded tissues were serially sliced to a thickness of 3 μm. Next, the slices were floated on a spreader of 40 °C warm water to spread the tissues. After, the tissues were fished up with slides, baked in an oven at 60 °C, dried, and removed for storage at room temperature. Immunohistochemistry was performed as previously described [26]. Briefly, the slides were washed in PBS at 37 °C for 30 min and closed in PBS containing 10% donkey serum, 1% BSA (Bovine Serum Albumin), and 0.3% Triton X-100 at 24 °C for 2 h. The slides were incubated with primary antibody overnight at 4 °C, washed in PBS containing 0.3% Triton X-100 before incubation, and then incubated with secondary antibody at room temperature for 2 h. Lastly, the slides were washed in PBS containing 0.3% Triton X-100, and sections were treated with Hoechst solution (10 mg/mL) for 5 min to stain cell nuclei. The working concentrations of the antibodies used were as follows: MIER3 antibody, 5 mg/mL; and Hoechst 33,342 (Cat. No. C1025; Beyotime Biotechnology Co., Ltd., Shanghai, China), 1:100.

### 4.9. Construction of MIER3–W Overexpression Vector

Based on the coding region of chicken *MIER3–W* (NCBI gene ID: 426615, accession: XM_015300440.3), the *MIER3–W* overexpression plasmid was designed by Jima Genetics Co., Ltd., Shanghai, China. The plasmid was lentivirally packaged by Kinkai Rui Biological Engineering Co., Ltd., Wuhan, China.

### 4.10. Isolation and Processing of Chicken Embryo Gonadal Cells

On E9.5 (right after sex determination, early stage of gonadal development, n = 6), cells from the female and male gonads were isolated from chicken embryos using collagenase (Cat. No. 17104019; Gibco). Briefly, E9.5 chicken embryo gonads were collected in 1.5 mL centrifuge tubes with PBS. After aspirating and discarding PBS, 500 μL of pre-warmed collagenase was added, and the tissues were pipetted and heated at 37 °C for 30 s. This step was repeated until the gonads became invisible. Subsequently, 800 μL of 10% bovine serum (FBS)-Dulbecco’s Modified Eagle Medium (DMEM) was added to the digestion solution, mixed by pipetting, and centrifuged at 700× *g* for 5 min at room temperature. After aspirating and discarding the supernatant, we added 1 mL of PBS to resuspend and wash the cells, which were centrifuged at 700× *g* for 5 min at room temperature. Next, the supernatant was discarded, and 1 mL of complete medium (DMEM with 10% fetal bovine serum, 1% penicillin-streptomycin solution (100 IU/mL), and 10 μL EGF (20 ng/mL) was added, and the mixture was filtered into a 50 mL conical tube using 70 μm nylon mesh. After counting the cells, a complete medium was added to plate and culture the cells at 1 × 10^6^ cells/well. The cells were then transfected with *MIER3* overexpression plasmid and empty vector plasmid at 70% cell fusion using Lipofectamine 2000 (Cat. No. 11668-027, Invitrogen, Carlsbad, CA, USA). Briefly, 4 μL of Lipofectamine 2000, *MIER3* overexpression plasmid and empty vector plasmid were diluted with 96 μL Opti-MEM. Next, the diluted Lipofectamine 2000 was mixed with diluted *MIER3* overexpression plasmid and empty vector and left to stand at room temperature for 23 min. The mixture was then added to the cultured cells. After 48 h of transfection, the cells were washed twice with PBS and then collected with 1 mL of TRIzol reagent or 100 μL of RIPA lysate and subjected to phenotypic assays.

### 4.11. Transcriptome Sequencing

Total RNA was extracted from cells transfected with *MIER3* overexpression vector or empty vector (n = 3) and sent to Personal Biotechnology Co., Ltd. for transcriptome sequencing. The quantity and quality of total RNA samples were analyzed with a Qubit 2.0 (ThermoFisher Scientific Inc., Waltham, MA, USA) and Agilent 2100 Bioanalyzer (Agilent Technologies, Santa Clara, CA, USA), respectively. The qualified total RNA samples were subsequently used to construct cDNA libraries by reverse transcription. The quality of this library was checked and sequenced using synthetic technology on the Illumina high-throughput sequencing platform HiSeq X-ten. Clean data with a score of ≥Q30 were selected for further analysis based on the calculated Phred scores of a large amount of raw data. The clean data were annotated using the HISAT2 system by comparing the reference genome sequence of chickens. Individual genes were assembled by comparing the reads using the StringTie program. Next, the expression of individual genes was calculated by normalizing the fragment per kilobase transcript/fragment per million map maps (FPKM) reads. Differential gene expression analysis was performed using DESeq (1.20.0) to screen for differentially expressed genes under the following conditions: expression difference fold |log2FoldChange| > 1 and significance *p* < 0.05. Gene ontology (GO) enrichment analysis was performed using top GO, and the *p* was calculated by the hypergeometric distribution method (the criterion for significant enrichment was *p* < 0.05). GO terms with significant enrichment of differential genes were identified to determine the main biological functions of the differential genes. KEGG pathway enrichment analysis was performed using clusterProfiler (3.4.4) software, focusing on significantly enriched pathways with *p* < 0.05.

### 4.12. Cell Phenotype

The proliferative ability of chicken embryonic gonadal cells was assessed using CCK-8 (Cat. No. A311-02; Vazyme Biotech Co., Ltd.) according to the manufacturer’s instructions. The cells transfected with *MIER3* overexpression vector or negative control vector (n = 4) were inoculated in 96-well plates (100 μL/well) at 2 × 10^3^/well, and five replicate wells were set up for each group. The cells were incubated for 12, 24, 48, 72, and 96 h with 10 μL CCK-8 in each well, and the absorbance (A450 nm) value of each well was measured at 450 nm by enzyme marker after incubation for 2 h at 37 °C. The background A450 nm value was removed by zeroing the blank control group, and the cell growth curve was statistically analyzed and plotted. Flow cytometry was used to detect apoptosis in chicken embryonic gonadal cells, and *MIER3* overexpression vector or negative control vector was transfected according to the instructions of the cell Apoptosis Kit (Cat. No. A211; Novozyme Biotechnology Co., Ltd. Nanjing, China). Untreated cells were used as a negative control. Additionally, for flow cytometry detection, annexin V-FITC and propidium iodide (PI) monochromes were required to regulate compensation. After 48 h, cells in each group were digested with trypsin without EDTA and collected. Next, the cells were washed twice with precooled PBS, and then 100 µL of 1× binding buffer was added and gently blown into a single-cell suspension. In addition, annexin V-FITC and PI staining were performed on the samples for flow cytometry detection to regulate compensation. After transfection with the *MIER3* overexpression vector for 48 h, the cells in each group were digested with trypsin without EDTA and collected. The cells were washed twice with precooled PBS and then gently blown into a single-cell suspension with 100 µL of 1× binding buffer. Subsequently, 5 μL Annexin V-FITC and 5 μL PI staining solution were added at room temperature (20–25 °C) for 10 min, followed by 400 μL of 1× binding buffer, and the mixture was blended gently. The samples were analyzed by flow cytometry within 1 h of staining, and the apoptosis distribution map was statistically analyzed and drawn. According to the manufacturer’s instructions for the kit (Cat. No. C1052; Beyotime Biotechnology Co., Ltd., Shanghai, China), the cell cycle of chicken embryonic gonad cells was detected by flow cytometry. Briefly, the cells in each group were collected by trypsin digestion after transfection with the *MIER3* overexpression and negative control vectors for 48 h. After 1 mL of precooled PBS was added to resuspended cells, 1 mL of 70% ethanol was added to fix the cells at 4 °C for 12 h, and then precooled 1 mL PBS was added to resuspended cells. Lastly, 0.5 mL of PI staining solution was added to the pre-prepared. After the resuspended cells were precipitated and incubated at 37 °C in the dark for 30 min, the cell cycle distribution was detected by flow cytometry.

### 4.13. Statistical Analysis

All values are expressed as the mean ± standard deviation. SPSS (version 16.0, SPSS China, Shanghai, China) was used to perform a Student’s *t*-test or one-way analysis of variance (Duncan’s test) to test for statistical significance differences between or among different groups. Statistical significance was set at *p* < 0.05.

## 5. Conclusions

We investigated the spatiotemporal expression patterns and potential functions of chicken *MIER3* in the gonadal development of chicken embryos. We found that *MIER3* gene expression was associated with the gonadal phenotype and may be involved in female gonadal development, supporting the idea that the W chromosome gene may be involved in the complete function of female gonads. However, as the *MIER3* gene expression level in female gonads is significantly higher than that in males, it is necessary to interfere with *MIER3* in female gonadal cells to further confirm its downstream pathways and genes. We tried investigating this process but failed due to the low interference efficiency. Future studies on how *MIER3* promotes female gonadal development through downstream genes may facilitate further understanding of chicken embryonic gonad development.

## Figures and Tables

**Figure 1 ijms-24-08891-f001:**
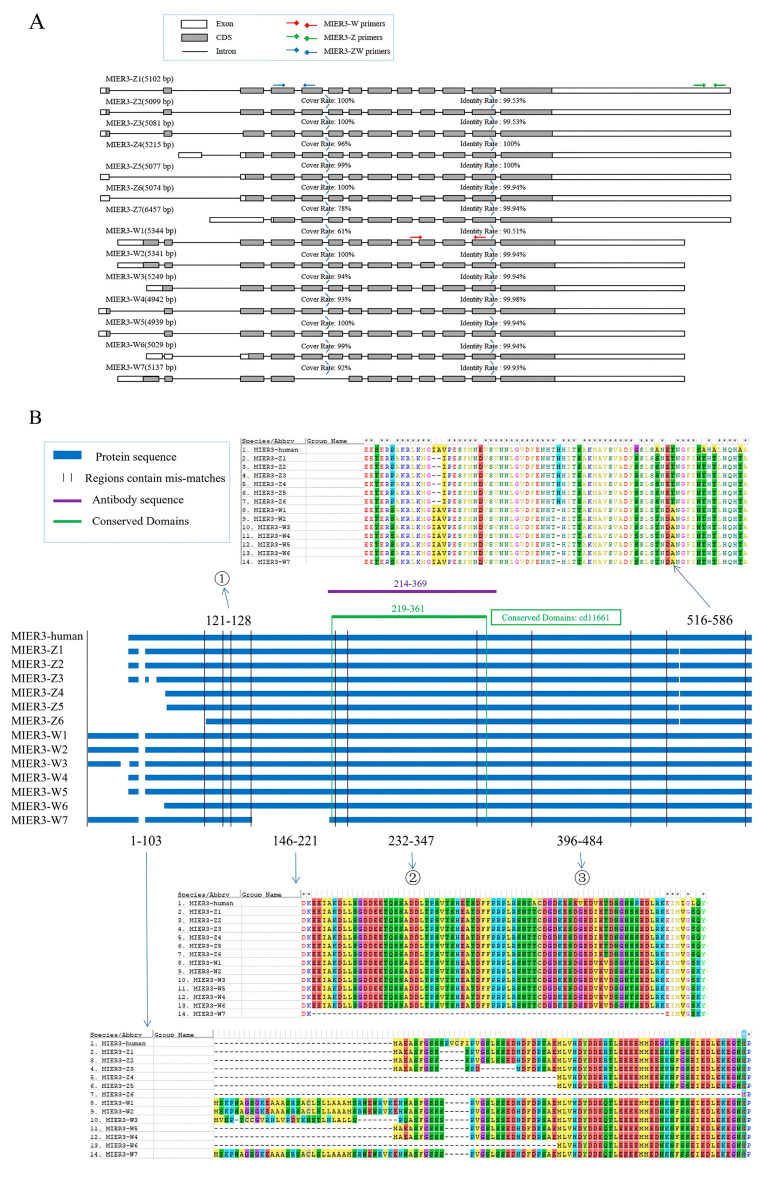
Schematic representation of the mRNA and protein sequences of MIER3 in chicken. (**A**) Comparison of the mRNA structures of *MIER3-W* (isoforms 1–7) and *MIER3-Z* (isoforms 1–7) in chicken. Coding sequences (CDS), introns, and qPCR primers specific for *MIER3-W*, *MIER3-Z,* and their shared regions are indicated. (**B**) Comparison of protein sequences of human MIER3 (isoform 1), chicken MIER3-Z, and chicken MIER3-W. Mismatched sequences are indicated by colored backgrounds, and conserved structural domains and antibody binding regions are also indicated. ①, ②, and ③ are detailed in Appendix A.

**Figure 2 ijms-24-08891-f002:**
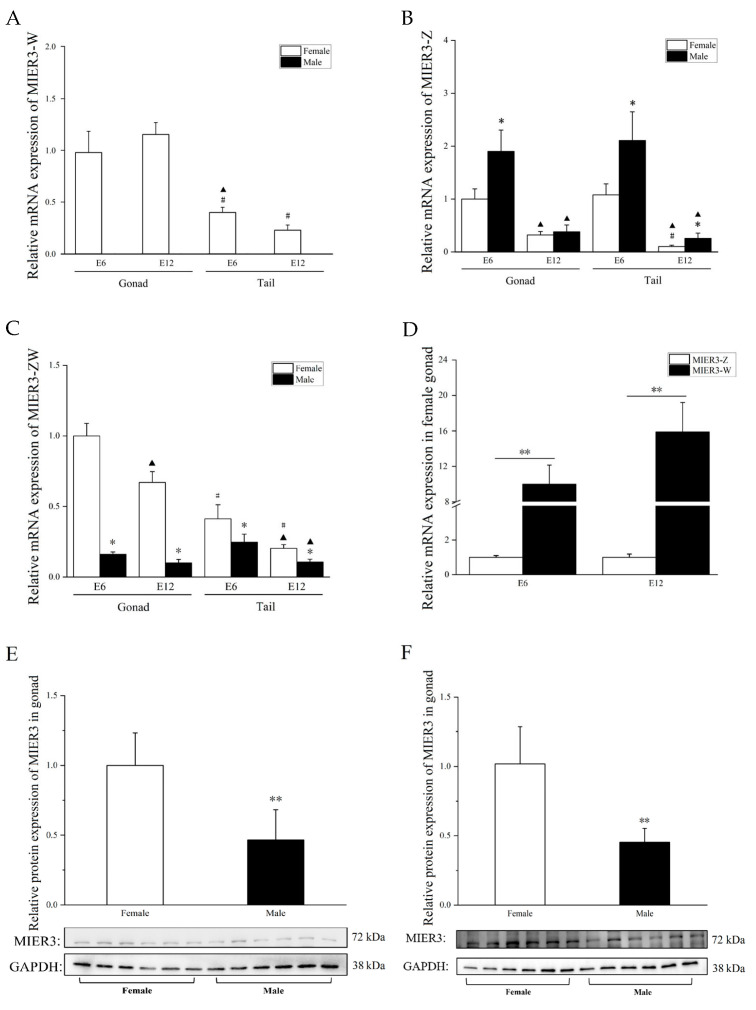
Spatiotemporal expression of *MIER3* gene. Relative expression of *MIER3–W* (**A**), *MIER3–Z* (**B**), and *MIER3–ZW* (**C**) in chicken embryonic gonadal tissue and non-gonadal tissue (tail) by qPCR; Relative expression of *MIER3–W* and *MIER3–Z* in female gonads by qPCR (**D**). (**E**,**F**) Protein levels of *MIER3* and *GAPDH* genes in E6 (**E**) and E12 (**F**) gonads in chickens (n = 6). F = female, M = male. In (**A**–**C**), * represents *p* < 0.05 between female and male in the same tissue at the same period, ▲ represents *p* < 0.05 between E6 and E12 in the same tissue of the same sex, and # represents *p* < 0.05 between gonads and tail tissue of the same sex at the same period. In (**D**–**F)**, * and ** represent *p* < 0.05 and *p* < 0.01, respectively. Moreover, all data are expressed as mean ± standard deviation.

**Figure 3 ijms-24-08891-f003:**
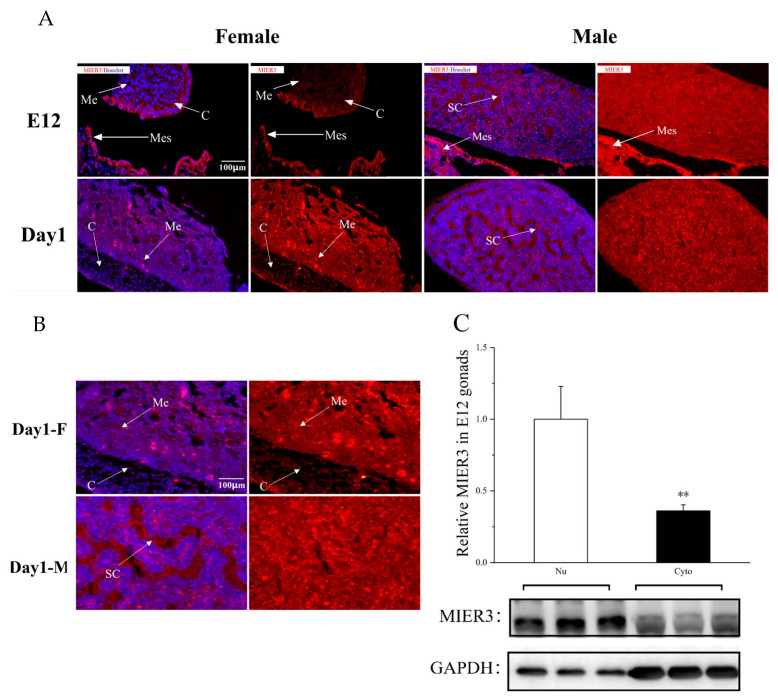
Localization of MIER3 protein in the chicken embryonic gonads. (**A**) Chicken MIER3 (red) and Hoechst (blue) expression in E12 and Day 1 gonad sections. Scale bar of all images is 100 μm. (**B**) Protein expression of MIER3 (red) and Hoechst (blue) in Day 1 gonadal sections: F = female; M = male; C = cortex; Me = medulla; Mes = Mesonephros; and SC = sex cord. All images are at a scale of 100 μm. (**C**) Relative protein expression of MIER3 to GAPDH in the nucleus and cytoplasm of E12 female gonads (n = 3). Nu = nuclei, Cyto = cytoplasm; M = male, F = female; ** represent *p* < 0.01, respectively, and all data are presented as mean ± standard deviation.

**Figure 4 ijms-24-08891-f004:**
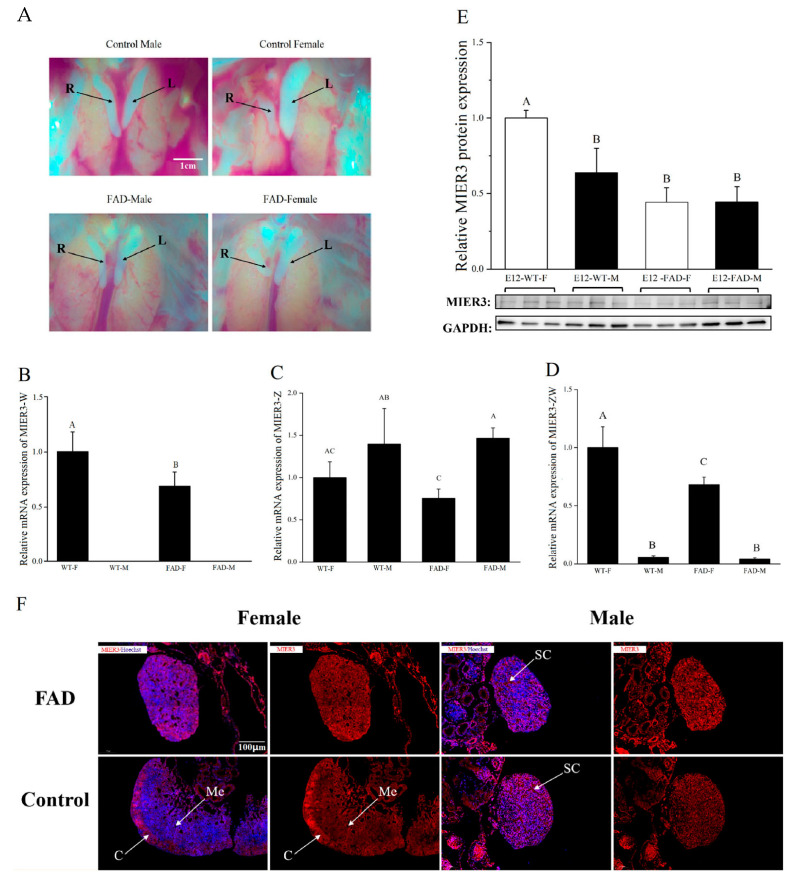
*MIER3* expression in the gonads of fatraconazole (FAD)-treated embryos. (**A**) Gross morphology of the gonads of E12 control and FAD-treated groups. R: Right side gonad; L: Left side gonad. (**B**–**D**) Relative expression of *MIER3* gene mRNA in the gonads of control (WT) and FAD-treated male and female embryos detected by qPCR: *MIER3–W* (**B**), *MIER3–Z* (**C**), and *MIER3–ZW* (**D**). (**E**) Relative expression of MIER3 protein in the gonads of control (WT) and FAD-treated E12 chicken embryos. F = female, M = male; E12 = Embryonic day 12; and WT = control. FAD refers to fadrozol injection. (**F**) Localization of MIER3 (red) and Hoechst (blue) in the gonads of FAD-treated embryos; C = cortex; Me = medulla; and SC = sex cord. Scale bar for all images is 100 μm. mRNA and protein expression were calculated relative to HMBS and GAPDH, respectively. Different letter columns are significantly different (*p* < 0.05), and all data are presented as mean ± standard deviation.

**Figure 5 ijms-24-08891-f005:**
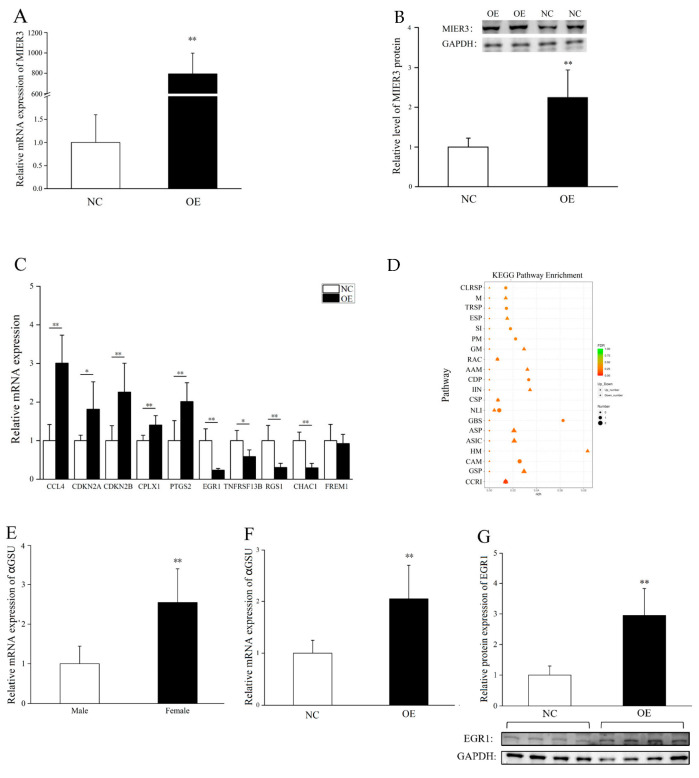
Effect of *MIER3–W* overexpression on downstream genes in gonadal cells. (**A**,**B**) Expression of *MIER3* gene was detected by qPCR (**A**) and WB (**B**) in male gonadal cells transfected with overexpression vectors or empty vectors (n = 6 and n = 2, respectively); (**C**) Verification of differentially expressed genes in control and *MIER3* overexpression groups in chicken embryonic gonadal cells; (**D**) Dot plots show the KEGG enrichment analysis results. Note: Horizontal coordinates indicate the ratio of the number of differentially expressed genes annotated on a specific KEGG pathway to the total number of differentially expressed genes annotated, and vertical coordinates indicate the KEGG pathway. The color indicates the adjusted *p*-value (padj), the size of the dot indicates the number of upregulated differentially expressed genes, and the size of the triangle indicates the number of downregulated differentially expressed genes. Experimental groups for the DEG screen were gonadal cells with *MIER3* overexpression vector, not empty vector (n = 3). CLRSP = C-type lectin receptor signaling pathway; M = Melanogenesis; TRSP = Toll-like receptor signaling pathway; ESP = ErbB signaling pathway; SI = Salmonella infection; PM = Pyrimidine metabolism; GM = Glutathione metabolism; RAC = Regulation of actin cytoskeleton; AAM = Arachidonic acid metabolism; CDP = Cytosolic DNA-sensing pathway; IIN = Intestinal immune network for lgA production; CSP = Calcium signaling pathway; NLI = Neuroactive ligand-receptor interaction; GBS = Glycosaminoglycan biosynthesis-heparan sulfate/heparin; ASP = Apelin signaling pathway; ASIC = Adrenergic signaling in cardiomyocytes; HM = Histidine metabolism; CAM = Cell adhesion molecules; GSP = GnRH signaling pathway; and CCRI = Cytokine-cytokine receptor interaction. (**E**) Relative expression of *αGSU* in E6 male and female chicken embryonic gonads (n = 6); (**F**) Relative expression of *αGSU* in male gonadal cells transfected with *MIER3* overexpression vectors or empty vectors (n = 6); (**G**) Protein levels of EGR1 in male gonadal cells transfected with *MIER3* overexpression vectors or empty vectors (n = 4). “OE” refers to the overexpression group, and “NC” refers to the negative control group. * and ** represent *p* < 0.05 and *p* < 0.01. All data are presented as mean ± standard deviation.

**Figure 6 ijms-24-08891-f006:**
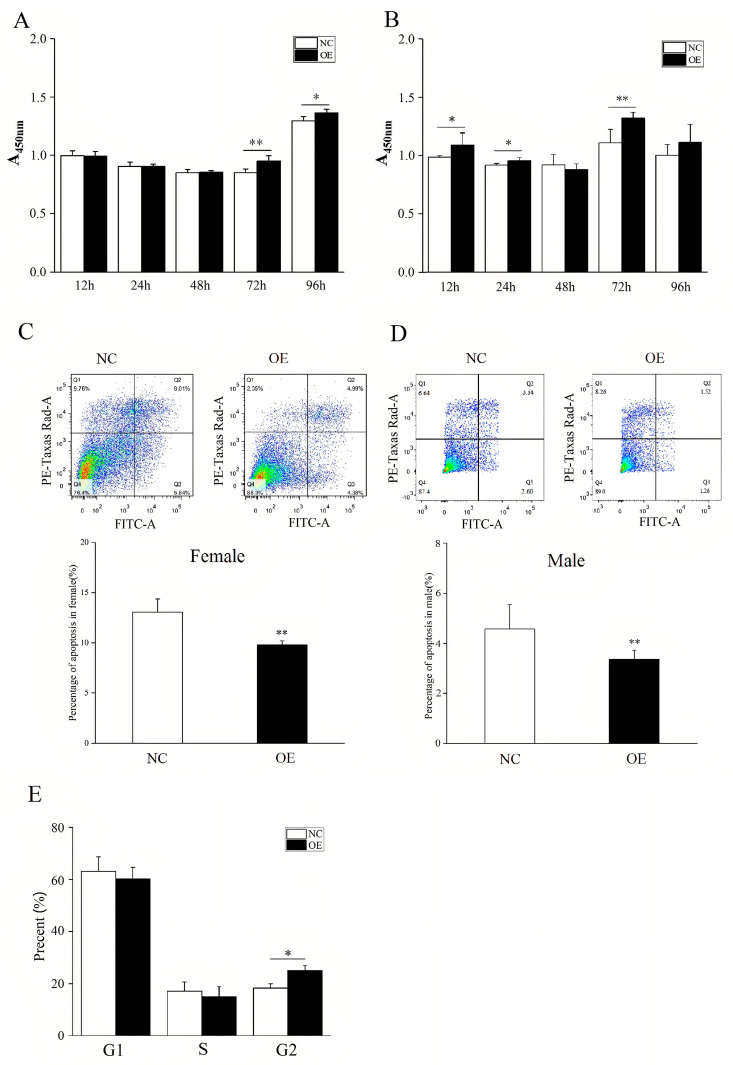
Effect of *MIER3* overexpression on gonadal cell phenotype. Effect of overexpression of *MIER3* gene on cell phenotype in E9.5 gonadal cells: (**A**,**B**) Cell proliferation (Female–(**A**), Male–(**B**)); (**C**,**D**) apoptosis (Female–(**C**); Male–(**D**)); (**E**) cell cycle (Male–(**E**)), n = 4. “OE” refers to the overexpression group, and “NC” refers to the negative control group. * and ** represent *p* < 0.05 and *p* < 0.01, respectively. All data are presented as mean ± standard deviation.

## Data Availability

The data presented in this study are available on request from the corresponding author.

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
