# Peer review of "Preliminary Study on Expression and Function of the Chicken W Chromosome Gene *MIER3* in Embryonic Gonads"

_ijms, 2023, doi:10.3390/ijms24108891_

Round 1
Reviewer 1 Report
The study is on an interesting topic and holds promise helping to understand a gene that may function in sex determination in chicken as a model organism. The authors did a substantial amount of experimental work. However, there are some problems that need to be addressed.
Conceptual comments
The study starts by discussing sex determination in birds regarding the DMRT1 gene, but then does not justify sufficiently in the introduction why the genes on the W chromosome should be studied, in light of other work on sex determination.
There is no rationale put forth to justify how the mier3 gene could function in sex determination given that it is expressed in the germ line tissues of both sexes. Even if it is expressed several fold more in the female tissue, it is not clear a priori how such a gene could function exclusively for one sex. Also, why might the gene be located on and expressed by both the Z and W chromosomes? Is the W chromosome copy a degenerative copy? (like some Y chromosomes that are thought to be degenerative X chromosomes and thus losing genes)
Why do the authors compare mier3 in chicken to the human ortholog? There is no rationale given as to how it relates to chicken sex determination. Is there any work from other studies suggesting mier3 function in humans? Is human meir3 important for human sex determination, especially in light of the fact that it is not sex-linked in this species? This type of information would add important context.
The introduction is deficient in background information on the W genes, as well as the CASI model that has been put forth to explain sex determination in birds.
Experimental comments
Regarding fig1a, what are the predicted changes in the ORF (encoded protein) between male and female isoforms? Which isoforms are differentially expressed between the sexes? The RT-qPCR analysis seems to detect the common part of these isoforms between the sexes and so it does not give isoform resolution, and that would help to determine if different proteins are produced between the sexes.
In the Western blot analysis, the protein amounts do not appear to be different substantially between the sexes.
The authors should state explicitly what FAD treatment does (this may not be obvious to a non-expert)
The authors need to clarify whether aromatase induction (by FAD treatment) is downstream of mier3 expression.
The tissue level analysis of MIER3 localization in figures 3 and 4 is very difficult to see any resolution at the cellular level. The authors could provide high magnification images so the reader can see at the cellular level. Also, panels a and b of figure 3 and panel f in figure 4 do not seem to include non-gonadal tissues, which would be an important control.
The authors rely on overexpression of mier3 to determine possible downstream pathway targets of sex determination. However, it is entirely possible, even likely, that overexpression would lead to inappropriate off-target effects that do not reflect the biology of the pathway, and the authors have no way of distinguishing among those effects that are in the pathway vs. those that are not. A standard genetic approach is to look at effects due to reduction of a gene of interest, such as through a mutation or through RNA interference, morpholine treatment, etc. It seems like the authors could use the viral delivery method to express hairpin RNA against their target gene to achieve knockdown, and then look for effects on downstream genes. This would be a more credible means of identifying downstream targets than overexpression.
The authors should describe to the reader what the pattern is for mier3 misexpression. Are the relationships of all of the affected genes sensible based on what the authors suspect meir3 does in its function? Or are the affected genes rather unrelated in their functional groups?
What might be the relationship(s) if any of meir3 to DMRT1?
Lines 125-127: 1.4-fold and 1.3-fold do not seem to be substantially different.
Technical comments
There is no information in the methods section on the phylogenetic analysis used in the study.
There are many grammatical errors and mis-spellings throughout the manuscript that need to be addressed.
There are multiple statements across the manuscript that need references or citations. For example, line 57 “MIER3 has been suggested to be involved in the sex determination or development of chicken gonads.” Was this suggestion from a previous study of mier3?
The statement in lines 91-92 does not make sense.
Line 284: “…have a more complicated mechanism.” More complicated that what? Please be specific.
The characters and labels of most of the figures are unreadable because they are too small. For example, the letters of the protein sequences in Figure 1 are unreadable. Another example is panel C of figure 5. This problem is applicable to nearly all panels/figures in the manuscript.
Author Response
Dear editor and reviewers:
Thank you and the reviewers very much for the opportunity and constructive critiques to improve this manuscript. We have carefully considered all the comments and suggestions and made the corresponding revisions (highlighted in yellow). The point-by-point responses are shown below with the comments in bold font and the responses in regular font.
The study is on an interesting topic and holds promise helping to understand a gene that may function in sex determination in chicken as a model organism. The authors did a substantial amount of experimental work. However, there are some problems that need to be addressed.
Conceptual comments
The study starts by discussing sex determination in birds regarding the DMRT1 gene, but then does not justify sufficiently in the introduction why the genes on the W chromosome should be studied, in light of other work on sex determination.
Response: Thank you very much for your comment. The rationale we think W chromosome should be studied was described in line 40 to 47 (The heterozygous knockout of the Z chromosome gene DMRT1 reportedly allows complete development of ovaries in ZZ-type chicken embryos, indicating that DMRT1 is a good candidate gene for chicken sex determination. However, these Z+Z- individuals did not develop into functional females (unable to lay eggs) when they reached adulthood and remained male in appearance and body type. This suggests that chickens may have more complex mechanisms of gonadal development (especially for female) other than the sole control of Z chromosomal genes’ dosage effect, such as the synergistic effects of W chromosomal genes.
Actually, we started this study with the presumption that MIER3 might be involved in female gonad sex determination or development (based on preliminary mRNA expression data), and it turned out that MIER3 was associated with ovary development other than sex determination (data not shown in the manuscript, see below for details).
There is no rationale put forth to justify how the MIER3 gene could function in sex determination given that it is expressed in the germ line tissues of both sexes. Even if it is expressed several fold more in the female tissue, it is not clear a priori how such a gene could function exclusively for one sex.
Response: Thank you for your comment and we agree with that. Actually, this study mainly focused on the role of MIER3 in gonadal development. We revised our presumption in the introduction section accordingly (line 54-55).
Also, why might the gene be located on and expressed by both the Z and W chromosomes? Is the W chromosome copy a degenerative copy? (like some Y chromosomes that are thought to be degenerative X chromosomes and thus losing genes)
Response: Yes, as far as we know, all protein coding W genes have their homologous copies on the Z chromosome, including the MIER3 gene. As the W chromosome is much smaller that the Z chromosome, it’s normally considered as a degenerative copy of the Z chromosome.
Why do the authors compare MIER3 in chicken to the human ortholog?
Response: As there is very few study about chicken MIER3, and studies in humans MIER3 showed its relationship with cell proliferation, so we compared chicken MIER3 to human MIER3 sequences, to show the possibility that they might have conserved function.
There is no rationale given as to how it relates to chicken sex determination.
Response: We agree. As we replied above, we used to assume that MIER3 might be involved in sex determination or ovary development, but it seems that MIER3 has no effect on sex determination. As we constructed an overexpression lentivirus vector of MIER3 gene and injected it to E2.5 chicken embryo. After successful injection, the gonadal phenotype was not changed. Please see the Figures below. Figure A shows the phenotype of gonads after the injection of MIER3 overexpression lentivirus vector, and Figure B shows the expression of MIER3 in E6, E9, E12 and E18 after successful injection. So we revised our presumption in the introduction section (line 54-55).
A
B
Is there any work from other studies suggesting MIER3 function in humans?
Response: Yes. Some studies have proved that MIER3 is related to breast cancer and rectal cancer in humans. MIER3 can significantly inhibit the proliferation and invasion of colorectal cancer cell lines; The research in breast cancer found that MIER3 is a susceptible gene of breast cancer, and its expression in breast cancer is significantly higher than that in normal breast tissue, suggesting that MIER3 is closely related to the occurrence and development of breast cancer (line 301-305).
Is human MIER3 important for human sex determination, especially in light of the fact that it is not sex-linked in this species? This type of information would add important context.
Response: No, there is no report about the relationship between human MIER3 gene and sex, which support the notion that chicken MIER3 may not be involved in sex determination.
The introduction is deficient in background information on the W genes, as well as the CASI model that has been put forth to explain sex determination in birds.
Response: Thank you for your suggestion. We have added the information of the W chromosome gene in lines 35-38 of the article. CASI is an important mechanism in chicken sex determination, especially for the non-gonadal tissues. However, as we would like to focus on the involvement of W genes in female gonadal development, and we do not see the logical relationship between W genes and CASI, so we did not include CASI in the introduction.
Experimental comments
Regarding Fig1a, what are the predicted changes in the ORF (encoded protein) between male and female isoforms? Which isoforms are differentially expressed between the sexes?
Response: As Figure 1A is a schematic diagram, detailed difference of ORF between different isoforms can not be shown. Instead, it shows the Structural differences which we think would be more informative. As to the question “which isoforms are differentially expressed between the sexes”, we can not tell because we designed the primers according to the common region of Z or W isoforms, respectively, and the expression data showed the overall expression of all the Z or W isoforms (except for unkown isoforms).
In the Western blot analysis, the protein amounts do not appear to be different substantially between the sexes.
Response: It’s true that the protein expression does not show as much difference as mRNA expression. But the difference is significant by calculating with software.
The authors should state explicitly what FAD treatment does (this may not be obvious to a non-expert).
Response: Thank you for your very good advice. We have added the explanation of FAD at its first appearance in line 58 - 59.
The authors need to clarify whether aromatase induction (by FAD treatment) is downstream of MIER3 expression.
Response: Thanks for your suggestion. We believe that aromatase induction is upstream of MIER3 expression, as FAD treatment decreased the expression of MIER3 while MIER3 overexpression had no effect on E2 pathway genes (FOXL2, AROM and ERα, data not shown). We have added the discussion in line 354-357.
The tissue level analysis of MIER3 localization in Figures 3 and 4 is very difficult to see any resolution at the cellular level. The authors could provide high magnification images so the reader can see at the cellular level. Also, panels a and b of Figure 3 and panel f in Figure 4 do not seem to include non-gonadal tissues, which would be an important control.
Response: Thank you very much for your suggestions. It’s true that detailed MIER3 localization is difficult to see from Figure 3 and 4 as we do not have confocal images. These Figures can only show the overall distribution of MIER3 in gonads and MIER3 is likely to express both in the nucleus and cytoplasm. And to prove this, we isolated nucleus and cytoplasm to detect the protein expression of MIER3 (Figure 3C). And we replaced the image of E12 female in Figure 3A with a higher magnification image.
The authors rely on overexpression of MIER3 to determine possible downstream pathway targets of sex determination. However, it is entirely possible, even likely, that overexpression would lead to inappropriate off-target effects that do not reflect the biology of the pathway, and the authors have no way of distinguishing among those effects that are in the pathway vs. those that are not. A standard genetic approach is to look at effects due to reduction of a gene of interest, such as through a mutation or through RNA interference, morpholine treatment, etc. It seems like the authors could use the viral delivery method to express hairpin RNA against their target gene to achieve knockdown, and then look for effects on downstream genes. This would be a more credible means of identifying downstream targets than overexpression.
Response: Thank you for your very good advice. We have done RNA interference in gonadal cells with siRNA, but the interference effect was not ideal. We have not constructed a virus to interfere with MIER3. This is a very good proposal for further study of MIER3. However, in the present study, in order to further validate the downstream genes, we have verified the expression of some downstream genes between female and male gonads, and most of them are consistent with the transcriptome sequencing results (higher in female gonads than male gonads).
The authors should describe to the reader what the pattern is for MIER3 misexpression. Are the relationships of all of the affected genes sensible based on what the authors suspect MIER3 does in its function? Or are the affected genes rather unrelated in their functional groups?
Response: Thanks you for your suggestion. However, we are afraid it’s hard to describe the pattern for MIER3 misexpression. Because from the KEGG pathway analysis, we could see that not all the functional groups are related to what we suspect, although we did get some related ones, as we mentioned in line 218-221 of the manuscript.
What might be the relationship(s) if any of MIER3 to DMRT1?
Response: That’s a great point. Actually, we did consider the relationship between MIER3 and DMRT1. Although we don’t have evidence about the effect of DMRT1 on MIER3, we knows that MIER3 overexpression could not affect DMRT1, neither in gonadal cells (our RNA-seq data) nor in vivo (embryo injection data, not shown in the manuscript). So we did not include this point in the manuscript.
Lines 125-127: 1.4-fold and 1.3-fold do not seem to be substantially different.
Response: Yes, it’s true that the fold-change is not substantial, but we confirmed that the difference is statistically significant.
Technical comments
There is no information in the methods section on the phylogenetic analysis used in the study.
Response: Thank you very much for your suggestions. It has been added in line 375 - 386 of the article.
There are many grammatical and spelling errors in the entire manuscript that need to be addressed.
Response: We have tried our best to improve the language and changes were marked yellow in the article.
There are multiple statements across the manuscript that need references or citations. For example, line 57 “MIER3 has been suggested to be involved in the sex determination or development of chicken gonads.” Was this suggestion from a previous study of MIER3?
Response: Thanks. For the sentence in line 57, which was our suggestion from our preliminary expression data, have now been revised. Other statements that we have added references were marked yellow.
The statement in lines 91-92 does not make sense.
Response: Thanks. We have deleted that statement.
Line 284: “…have a more complicated mechanism.” More complicated that what? Please be specific.
Response: Sorry for not making it clear. We have added this information in lines 282-283 of the manuscript.
The characters and labels of most of the Figures are unreadable because they are too small. For example, the letters of the protein sequences in Figure 1 are unreadable. Another example is panel C of Figure 5. This problem is applicable to nearly all panels/figures in the manuscript.
Response: Thank you very much for your suggestion. We have made the labels bigger.

Reviewer 2 Report
The standard showed that the expression of MIER3 is responsible for the development of the gonads. This is very important due to the connection of this achievement with the results of research with chicken breeding. The influence of the source, which gonads develop in the embryo, individual characteristics and sex of the female can be extracted for laying hens or for breeding male broilers, which result in higher slaughter value. The use of immunostaining to test for reversed sex will eliminate as many individuals from reproduction.
Add in "Materials and methods" how many embryos were analyzed?
The authors in the discussion cite few results of other researchers.
The end of the discussion section should be changed and discussed with other scientists. This part is is a statement to the research results of the reviewed work.
Due to the unfinished process of this research, we propose to change the title of the work to "Preliminary study on expression and function of the chicken W chromosome gene MIER3 in
embryonic gonads".
Author Response
Dear editor and reviewers:
Thank you and the reviewers very much for the opportunity and constructive critiques to improve this manuscript. We have carefully considered all the comments and suggestions and made the corresponding revisions (highlighted in yellow). The point-by-point responses are shown below with the comments in bold font and the responses in regular font.
The standard showed that the expression of MIER3 is responsible for the development of the gonads. This is very important due to the connection of this achievement with the results of research with chicken breeding. The influence of the source, which gonads develop in the embryo, individual characteristics and sex of the female can be extracted for laying hens or for breeding male broilers, which result in higher slaughter value. The use of immunostaining to test for reversed sex will eliminate as many individuals from reproduction.
Add in "Materials and methods" how many embryos were analyzed?
Response: Thank you very much for your suggestions. We have added the number of embryos to the materials and methods section (line 367, 392, 450, 473, 494).
The authors in the discussion cite few results of other researchers. The end of the discussion section should be changed and discussed with other scientists. This part is is a statement to the research results of the reviewed work.
Response: Thank you for your suggestions. We have cited more references to add to the introduction (line 35-38) or discussion (line 354-357).
Due to the unfinished process of this research, we propose to change the title of the work to "Preliminary study on expression and function of the chicken W chromosome gene MIER3 in embryonic gonads".
Response: Thank you very much for your suggestion. We have changed the title of the article.

Reviewer 3 Report
Lin et al. have used a number of standard molecular and cellular approaches (including but not limited to RT-qPCR, Western blotting, RNA-seq and FACS) to analyze the expression pattern of the MIER3 gene residing in sex Z and W chromosomes across different tissues during chicken embryogenesis. The particular attention was paid to the developing gonads in female and male embryos. Since the molecular mechanisms of sex determination in chicken are not fully understood, detailed characterization of activities of all W-linked genes (specific to females ZW, but not to males ZZ) is of fundamental interest and practical importance.
I have to notice very low quality of the presentation of the results in the manuscript. First, in the text, there are multiple references to Supplementary Figures and Tables, but none of them are shown/available. Showing the original images of blots is a good thing, but these images cannot substitute for the Supplementary Materials expected. As a result, it is simply not possible to judge some authors rationales and many of the conclusions drawn by the authors.
Next, the main Figures look like initial drafts. For example, microscopic images shown in Figure 3ab are not complete and the Western blot shown in Figure 5g is just the original (not modified) scan of a blot provided without any labelling. Some details in Figures are so small that cannot be unambiguously identified even after zooming in. Much more comments to Figures and their legends can be found in the enclosed PDF file.
The results on the MIER3 gene expression obtained by the RT-qPCR assay are quite limited considering the complex gene structure and the numbers of its mRNA isoforms. Firstly, the choice of the primers specific to MEIR3-W transcripts seems to be very doubtful according to Figure 1a. Secondly, RNA-seq, which allows distinguishing different transcript isoforms and, thus, providing much more knowledge on the biology of the MEIR3 gene, should be used in addition to RT-qPCR at least for some gonad samples.
The results of Western blot analysis and particularly the results of fluorescent immunostaining experiments are not solid, as the quality/specificity of the anti-MIER3 antibodies used is not known/studied (no such data are shown). Either it should be convincingly shown that these antibodies do not recognize any background (e.g., in the extracellular material) and/or the results obtained should be confirmed by using another antibodies against MIER3. Ideally, antibodies that recognized specific MIER3-W and MIER3-Z isoforms should be raised and used.
The list of references is very scarce and include completely not relevant papers (e.g., [11]). At the same time, a number of very relevant publications (e.g., including but not limited to DOI: 10.1080/10495398.2021.1935981 and 10.1186/gb-2013-14-3-r26) are not mentioned.
The standard rules of writing gene names in italic and protein names in regular font are completely ignored by the authors.
Additional minor comments that might be useful to improve the manuscript can be found in the enclosed PDF.

Author Response
Dear editor and reviewers:
Thank you and the reviewers very much for the opportunity and constructive critiques to improve this manuscript. We have carefully considered all the comments and suggestions and made the corresponding revisions (highlighted in yellow). The point-by-point responses are shown below with the comments in bold font and the responses in regular font.
Lin et al. have used a number of standard molecular and cellular approaches (including but not limited to RT-qPCR, Western blotting, RNA-seq and FACS) to analyze the expression pattern of the MIER3 gene residing in sex Z and W chromosomes across different tissues during chicken embryogenesis. The particular attention was paid to the developing gonads in female and male embryos. Since the molecular mechanisms of sex determination in chicken are not fully understood, detailed characterization of activities of all W-linked genes (specific to females ZW, but not to males ZZ) is of fundamental interest and practical importance.
I have to notice very low quality of the presentation of the results in the manuscript. First, in the text, there are multiple references to Supplementary Figures and Tables, but none of them are shown/available. Showing the original images of blots is a good thing, but these images cannot substitute for the Supplementary Materials expected. As a result, it is simply not possible to judge some authors rationales and many of the conclusions drawn by the authors.
Response: Sorry for the low quality of the Figures. We have revised the Figures to make them clearer. The Supplementary Figures and Tables were uploaded separately with the manuscript, and maybe they need to be download from somewhere in the system?
Next, the main Figures look like initial drafts. For example, microscopic images shown in Figure 3ab are not complete and the Western blot shown in Figure 5g is just the original (not modified) scan of a blot provided without any labelling. Some details in Figures are so small that cannot be unambiguously identified even after zooming in. Much more comments to Figures and their legends can be found in the enclosed PDF file.
Response: Sorry for the low quality of the Figures. We have revised the Figures.
The results on the MIER3 gene expression obtained by the RT-qPCR assay are quite limited considering the complex gene structure and the numbers of its mRNA isoforms. Firstly, the choice of the primers specific to MEIR3-W transcripts seems to be very doubtful according to Figure 1a. Secondly, RNA-seq, which allows distinguishing different transcript isoforms and, thus, providing much more knowledge on the biology of the MEIR3 gene, should be used in addition to RT-qPCR at least for some gonad samples.
Response: Thanks for the reviewer’s good advice. We actually tried to design specific primers for different isoforms, but the results were not satisfying (considering the specific sequence of each isoform will make the quality of primers very low, and reduce the efficiency of primers). For the present MIER3-W primer, its position on W isoform looks like having common region with Z isoform, according to Figure 1A. While indeed, the detail sequences are different in that region, which can not be shown from Figure 1A (as it’s just a schematic diagram). We have verified this pair of primers in ZZ and ZW samples and confirmed that this primer is working only in ZW samples.
RNA-seq would be a good way to distinguish the expression of different isoforms, which we will consider in the future. Thanks again for the very good suggestion.
The results of Western blot analysis and particularly the results of fluorescent immunostaining experiments are not solid, as the quality/specificity of the anti-MIER3 antibodies used is not known/studied (no such data are shown). Either it should be convincingly shown that these antibodies do not recognize any background (e.g., in the extracellular material) and/or the results obtained should be confirmed by using another antibodies against MIER3. Ideally, antibodies that recognized specific MIER3-W and MIER3-Z isoforms should be raised and used.
Response: Thank you for your comment. We have verified the MIER3 antibodies in the gonadal cells transfecteed with MIER3 overexpression vector or not., which showed a higher expression in the overexpression group (see figure bellow). Antibodies that recognized specific MIER3-W and MIER3-Z isoforms were too difficult to obtain. Actually, we raised the antibodies with MIER3-W, but it turned out that the antibodies recognize both W and Z isoforms (as we did not found suitable W specific sequences for antibody raising).
The list of references is very scarce and include completely not relevant papers (e.g., [11]). At the same time, a number of very relevant publications (e.g., including but not limited to DOI: 10.1080/10495398.2021.1935981 and 10.1186/gb-2013-14-3-r26) are not mentioned.
Response: Thank you very much for your suggestion. Previous reference 11 was a mistake and has been replaced with the correct one. Additional references (including your kind suggestions) have been added to the manuscript (line 35, 36, 38, 52, 276, 278 and 313).
The standard rules of writing gene names in italic and protein names in regular font are completely ignored by the authors.
Response: We have checked through the manuscript and revised where appropriate.
Additional minor comments that might be useful to improve the manuscript can be found in the enclosed PDF.
Line 15: What is the "combined expression of mRNA and protein?
Response: Sorry for the confusing description. What we were trying to state was “the overall expression of MIER3-Z and MIER3-W”. We have revised it in line 13.
Line 19-22: repeats and Contradictions!
Response: We have revised it.
Line 38-41: Ref?
Response: We have added the references.
Line 42: All genes in italic and all protein in regular font everywhere!
Response: We have checked through the manuscript and revised where appropriate.
Line 49: This manuscript is only about 1 gene, not about all 28 W chromosome genes!
Response: Sorry for the confusion. In our previous study, we analyzed the spatiotemporal expression of 28 genes on the W chromosome, based on which we selected MIER3 as a candidate gene. We have made this clear in line 47.
Line 54: TF: How is it known?
Response: We have added the reference.
Line 57: Data/Results supporting this statement?
Line 59: What are the "preliminary results"? Ref(s)?
Response: Figure A shows the expression levels of E6 and E12 chicken embryos in the gonads and tail tissues on the W chromosome, and candidate genes were then screened. Figure B shows the analysis of the expression levels of these candidate genes in the female and male gonads, and it is found that the expression level of MIER3 gene in the
female gonads is significantly higher than that in the male gonads.A
B
Line 61: Who and when suggested this hypothesis?
Response: This is a hypothesis based on our previous data, and now we deleted this sentence.
Line 74: According to the text above, only one human isoform was used for the comparison!
Response: Yes. Humans have four transcripts, and we chose the longest one.
Line 79: What are these domains? Ref(s)? Are they, for example, known to be in TFs?
Response: These areas are SANT, SANT-MTA3-like, and ELM2. These regions were found in the sequence we compared on the NCBI website. We have added the reference.
Line 80: Repetition! Combine this sentence with the previous one.?
Response: This two sentences are different. The latter one shows the common domain of all isoforms.
Line 84: What is the FIRST transcript? Is it always the longest one, which would be the most logical to use?
Response: Thanks for your suggestion. We aimed to use the longest one, and we checked for that and performed the analysis again.
Line 89: African lungfish is not an animal!
Response: Sorry, we don’t understand this.
Line 92: What do the cove(ar)rates mean?
Response: It means comparison rate between the previous transcript and the next transcript.
Line 92: What do circled numbers 1-3 mean?
Response: They referred to the regions where amino acid sequences of chicken MIER3-Z and MIER3-W differ from human MIER3. As we can not show all the regions in the figure, we put them to the Figure S1. We have indicated that in the figure legend.
Line 105: C is confusing, as it looks like that there is a third gene...
Response: We have revised C with ZW.
Line 122: What are the " left gonads"?
Response: It means the left side gonad.
Line 130: Antibody could be generated to regions specific to Z and W protein isoforms....
Response: Actually, we raised the antibodies with MIER3-W, but it turned out that the antibodies recognize both W and Z isoforms (as we did not found suitable W specific sequences for antibody raising).
Line 163: It looks like that only fragments of the expected full "red" panels are shown in both (a) and (b) panels.
Response: The previous figure had display problem. We have revised the figure.
Line 169: What does it mean? This contradicts with the scale bar definition.
Response: Thanks. We have revised it.
Line 174: It is necessary, at least briefly, describe HERE the approach used to reverse the sex of the embryonic gonads (including providing the appropriate references).
Response: Please refer to the FAD treatment of chicken embryos in Materials and Methods for details on the method of reversing the gender of embryos.
Line 190: What do "R" and "L" indicate to?
Response: Sorry, we didn't describe it clearly."R" refers to the right gonad and "L" refers to the left gonad. And we have added this into the legend.
Line 194: Where is "E12" used in panel (e)?
Response: The previous figure had display problem. We have revised the figure.
Line 197: The last two sentences of the legend do not belong to the panel (f), but rather to the panels (b)-(e), so their position is confusing.
Response: The previous figure had display problem. We have revised the figure.
Line 202: Also, the way used to demonstrate the significant difference between the samples analyzed is also very confusing. In the legend, 2 different approaches are mentioned, but * and ** are absent in the Figure!
Response: We have revised it in the legend.
Line 219: All gene names should be written in italic.
Response: We have revised them.
Line 245: What do "O" and "C" mean?
Response: "O" refers to the overexpression group, and "C" refers to the negative control group. We have corrected the logo in Figure 5B and it is consistent with the rest of the article.
Line 245: What do "NC" and "OE" mean?
Response: "OE" refers to the overexpression group, and "NC" refers to the negative control group.
Line 245: The font size used for the labels is too small.
Response: We have made the labels bigger.
Line 245: Some labels can't be read even after zoom in, particularly in (not labelled) panel with KEGG pathways.
Response: We have made the labels bigger.
Line 245: Wrong panel numbering starting from (d).
Response: The previous figure had display problem. We have revised the figure.
Line 245: What is this?
Response: The previous figure had display problem. We have revised the figure.
Line 270: What do "NC" and "OE" mean?
Response: "OE" refers to the overexpression group, and "NC" refers to the negative control group. We have added to the legend.
Line 270: What is "A450 nm"?
Response: The absorbance (A450nm) value of each well.
Line 270: Labels in FACS graphs are so small that almost unreadable.
Response: We have made the labels bigger.
Line 308: This is a wrong citation!
Response: We have revised the references here.
Line 333: A gene is always at the same position in the genome (not considering 3D genome organization in the nucleus), but the protein localization within a cell or a tissue can indeed vary...
Response: We have revised the sentence to make it clear (line 330-331).
Line 347: This could be just due to the background activity of the antibody used. How this option was excluded from the consideration?
Response: From the image of female gonads (Figure 3B), we can see that in the cortex, there are regions with very low level of MIER3 signal but with strong signal for dapi (nuclei).
Line 370: It is better to move this information up, to the appropriate position in the Results.
Response: Thank you for your suggestion. We have included this information in the results section (line 101-102).
Line 387: What is the CHD gene?
Response: Chromosome helicin genes (CHD) exist on the sexual chromosomes Z and W of birds. The CHD on the Z chromosome is denoted as CHD-Z, and the CHD on the W chromosome is denoted as CHD-W; By designing primers that amplify different lengths of product from Z and W copy, we can determine the genetic sex.
Line 389: What is its full name?
Response: Chromobox-helicase-DNA binding gene (line 396).
Line 391: How is it associated with the sex of chicken?
Response: As described above.
Line 398: Explain in more details. (4.5 RNA isolation, cDNA synthesis, and qPCR)
Response: Because E6 gonads are small, we collected 5 pairs of gonads to form a single pool, with a biological repeat of 6 for each tissue.
Line 549: ALL THIS SUPPLEMENTARY MATERIALS ARE ABSENT!
Response: The Supplementary Figures and Tables were uploaded separately with the manuscript, and maybe they need to be download from somewhere in the system?

Round 2
Reviewer 3 Report
Although the entire manuscript has been substantially improved, my two major concerns were not addressed by the authors. First, the necessity of using RNA-seq to distinguish different transcript isoforms (at least in some most interesting samples) is completely ignored. Second, no additional information about possible non-specific activity of the anti-MIER3 antibodies used was provided. Overexpression experiments cannot address this issue, they only demonstrate that the antibodies do recognize MIER3. The presence of regions in the cortex of female gonads with low level of putatively MIER3 signal but with strong signal for DAPI (shown in Figure 3B) also does not necessarily indicate that strong signals detected after immunostaining with the antibodies is exclusively/primarily due to the presence of the MIER3 protein. Once again, my question is whether these antibodies also recognize some other protein(s) in addition to MIER3 or not. Such protein(s) may occasionally have expression patterns in part similar to that of MIER3. So, this knowledge is crucial for the interpretation of (especially of) cytological results. Importantly, the images of original blots (e.g., those used to prepare Figures 2F and 3C) do suggest that these antibodies recognize something else besides the MIER3 protein. One standard approach to demonstrate the absence of unwanted background specificity of antibodies is to use tissues or (cultured) cells with knockdown/knockout for the target protein/gene as an input for Western blots and immunostaining experiments. Alternatively, authors can use completely different antibodies against MIER3 to confirm the currently presented results; it is quite possible that some of commercially available antibodies (raised even against non-chicken) MIER3 protein could work for this purpose (e.g., https://www.mybiosource.com/polyclonal-bovine-chicken-dog-horse-human-mouse-ovine-pig-rabbit-rat-antibody/mier3/9611900).
Minor comments
Manuscript (at least the available for me PDF file) contains many fused words and other mistyping errors. For example, in the manuscript title, the name of the MIER3 gene is not written in italic and “in” and “embryonic” are fused together. Besides multiple cases within the main text of the manuscript, the issue of fused words might also affect authors' names.
Lines 70-71, 301, 370: Genes/transcripts (DNA/RNA) do not have/contain amino acid sequences; instead they encode proteins.
Line 75: It is not clear what is meant under “enzyme catalysis”.
Lines 72-73, 76, 78-79, 85, 87, 294-295: The proteins are discussed, but the italic font is used.
Lines 133, 203-207, 231, 274, 389: Gene names should be written in italic.
Author Response
Dear editor and reviewers:
Thank you and the reviewers very much for the opportunity and constructive critiques to improve this manuscript. We have carefully considered all the comments and suggestions and made the corresponding revisions (highlighted in yellow). The point-by-point responses are shown below with the comments in bold font and the responses in regular font.
Although the entire manuscript has been substantially improved, my two major concerns were not addressed by the authors.
First, the necessity of using RNA-seq to distinguish different transcript isoforms (at least in some most interesting samples) is completely ignored.
Response: We apologize for not adding more information about different transcripts of MIER3-W in our previous revision. Although RNA-seq can distinguish different transcripts, its prediction of the expression of different transcripts is not very accurate due to the short read length. So instead, to clarify the expression of different transcripts of MIER3-W, we designed primers to amplify different transcripts according to their unique sequence (New Supplementary Table S1) and performed the assay of amplification efficiency (Supplementary Figure S2) and the assay of expression of different transcripts (Supplementary Figure S3) in E12 female gonads by qPCR, and the results showed that transcripts 3-6 were relatively highly expressed, while transcripts 1, 2 and 7 were very lowly expressed. In addition, since the sum of all transcripts expression is much lower than the total MIER3-W expression, it is possible that other unknown W transcripts are also present. We have added this description in the manuscript (line 111-118). In addition, please see below for the details of primer positions on the sequences.
MIER3W-127
Forward Primer:CCAAATATGGCGGAGGTG
Reward Primer:GACGTGGAAAGCCAGTCG
MIER3W-3
Forward Primer:GAGGCACCTTGTTCCAGACTA
Reward Primer:TTCCATCATTTCCTCCTCTTCG
MIER3W-45
Forward Primer:TGCTAGCGCTGTTGCTCAACC
Reward Primer:CCTGCCATAACTGGAATCGTG
MIER3W-6
Forward Primer: GAAGGAGCACTGGGTCAGTC
Reward Primer: TCGAACTCCCAAAGGATGCC
Second, no additional information about possible non-specific activity of the anti-MIER3 antibodies used was provided. Overexpression experiments cannot address this issue, they only demonstrate that the antibodies do recognize MIER3. The presence of regions in the cortex of female gonads with low level of putatively MIER3 signal but with strong signal for DAPI (shown in Figure 3B) also does not necessarily indicate that strong signals detected after immunostaining with the antibodies is exclusively/primarily due to the presence of the MIER3 protein. Once again, my question is whether these antibodies also recognize some other protein(s) in addition to MIER3 or not. Such protein(s) may occasionally have expression patterns in part similar to that of MIER3. So, this knowledge is crucial for the interpretation of (especially of) cytological results. Importantly, the images of original blots (e.g., those used to prepare Figures 2F and 3C) do suggest that these antibodies recognize something else besides the MIER3 protein. One standard approach to demonstrate the absence of unwanted background specificity of antibodies is to use tissues or (cultured) cells with knockdown/knockout for the target protein/gene as an input for Western blots and immunostaining experiments. Alternatively, authors can use completely different antibodies against MIER3 to confirm the currently presented results; it is quite possible that some of commercially available antibodies (raised even against non-chicken) MIER3 protein could work for this purpose (e.g., https://www.mybiosource.com/polyclonal-bovine-chicken-dog-horse-human-mouse-ovine-pig-rabbit-rat-antibody/mier3/9611900).
Response: Thanks for your suggestion, actually, we have purchased several commercial antibodies, such as MIER3 antibodies from Beijing Boorsen Biotechnology Co., Ltd. (product number: bs-18939R), MIER3 antibodies from Thermo Fisher Scientific Co., Ltd. (product number: PA5-31879), and MIER3 antibodies from Jiangsu Qinke Biological Research Center Co., Ltd. (product number: DF13150), but none of these antibodies gave ideal bands (as shown in the following figures). That’s why we made antibodies against chicken MIER3-W. We were about to validate the antibodies using knockdown samples, but the effect of knockdown experiments in gonadal cells was not good. However, as we did get strong band at the right position, and it can be validated in the overexpression samples, we are confident that at least the antibodies we made can recognize MIER3 protein. In that case, we think the antibodies can be used to show the general localization of MIER3 in the gonad tissues.
Minor comments
Manuscript (at least the available for me PDF file) contains many fused words and other mistyping errors. For example, in the manuscript title, the name of the MIER3 gene is not written in italic and “in” and “embryonic” are fused together. Besides multiple cases within the main text of the manuscript, the issue of fused words might also affect authors' names.
Lines 70-71, 301, 370: Genes/transcripts (DNA/RNA) do not have/contain amino acid sequences; instead they encode proteins.
Response: Thank you very much for your suggestions. We have made corresponding revisions (line 71,300,303).
Line 75: It is not clear what is meant under “enzyme catalysis”.
Response: Enzymatic catalysis can be regarded as a catalytic reaction between homogeneous and heterogeneous catalysis reactions. It can be seen as either the formation of intermediate compounds between reactants and enzymes, or the adsorption of reactants on the surface of enzymes before the reaction takes place.
Lines 72-73, 76, 78-79, 85, 87, 294-295: The proteins are discussed, but the italic font is used.
Response: Thank you very much for your suggestions. We have made corresponding revisions (line 71-79,87,306-308).
Lines 133, 203-207, 231, 274, 389: Gene names should be written in italic.
Response: Thank you very much for your suggestions. We have made corresponding revisions (line 143,243).

Round 3
Reviewer 3 Report
I think that the current version of manuscript can be accepted for publication after the following minor changes are introduced in the text.
Lines 300-301: “amino acid sequences encoded proteins of different transcripts on the Z and W copies” should be changed to “amino acid sequences encoded by different transcripts of the Z and W copies”
Line 301: both cases of “MIER3” should be written in italic, according to gene nomenclature (see details, for example, here: https://en.wikipedia.org/wiki/Gene_nomenclature).
Line 302: both cases of “MIER3” should be written in italic, according to gene nomenclature (see details, for example, here: https://en.wikipedia.org/wiki/Gene_nomenclature).
Lines 303-304: “Nevertheless, the amino acid sequences encoded proteins of human” should be changed to “Nevertheless, the amino acid sequences of human”.
Title of Fig. S2: “Standard curve” should be changed to “Standard curves”.
Title of Fig. S2: “transcripts of the MIER3 gene on W staining” should be changed to “transcripts of the MIER3-W gene” (“MIER3-W” should be written in italic).
Title of Fig. S3: “MIER3” should be written in italic, according to gene nomenclature (see details, for example, here: https://en.wikipedia.org/wiki/Gene_nomenclature).
Title of Fig. S4: “Evolutionary tree of MIER3 gene at amino acid sequence in different species” should be changed to “Evolutionary tree of MIER3 protein”.
Author Response
Dear editor and reviewers:
Thank you and the reviewers very much for the opportunity and constructive critiques to improve this manuscript. We have carefully considered all the comments and suggestions and made the corresponding revisions (highlighted in yellow). The point-by-point responses are shown below with the comments in bold font and the responses in regular font.
I think that the current version of manuscript can be accepted for publication after the following minor changes are introduced in the text.
Response: Thanks. We have made corresponding modifications in the manuscript according to your suggestions.
Lines 300-301: “amino acid sequences encoded proteins of different transcripts on the Z and W copies” should be changed to “amino acid sequences encoded by different transcripts of the Z and W copies”
Response: Thank you for your suggestion. We have made the corresponding revision (line 300-301).
Line 301: both cases of “MIER3” should be written in italic, according to gene nomenclature (see details, for example, here: https://en.wikipedia.org/wiki/Gene_nomenclature).
Response: Thank you for your suggestion. We have made the corresponding revisions (line 301).
Line 302: both cases of “MIER3” should be written in italic, according to gene nomenclature (see details, for example, here: https://en.wikipedia.org/wiki/Gene_nomenclature).
Response: Thank you for your suggestion. We have made the corresponding revisions (line 302).
Lines 303-304: “Nevertheless, the amino acid sequences encoded proteins of human” should be changed to “Nevertheless, the amino acid sequences of human”.
Response: Thank you for your suggestion. We have made the corresponding revision (Line303).
Title of Fig. S2: “Standard curve” should be changed to “Standard curves”.
Response: Thank you for your suggestion. We have made the corresponding revision in the supplementary file (Line12).
Title of Fig. S2: “transcripts of the MIER3 gene on W staining” should be changed to “transcripts of the MIER3-W gene” (“MIER3-W” should be written in italic).
Response: Thank you for your suggestion. We have made the corresponding revision in the supplementary file (Line14).
Title of Fig. S3: “MIER3” should be written in italic, according to gene nomenclature (see details, for example, here: https://en.wikipedia.org/wiki/Gene_nomenclature).
Response: Thank you for your suggestion. We have made the corresponding revision in the supplementary file (Line17).
Title of Fig. S4: “Evolutionary tree of MIER3 gene at amino acid sequence in different species” should be changed to “Evolutionary tree of MIER3 protein”.
Response: Thank you for your suggestion. We have made the corresponding revision in the supplementary file (Line18).
